# Preclinical Evaluation of the Antimicrobial-Immunomodulatory Dual Action of Xenohormetic Molecules against *Haemophilus influenzae* Respiratory Infection

**DOI:** 10.3390/biom9120891

**Published:** 2019-12-17

**Authors:** Ariadna Fernández-Calvet, Begoña Euba, Lucía Caballero, Roberto Díez-Martínez, Margarita Menéndez, Carlos Ortiz de Solórzano, José Leiva, Vicente Micol, Enrique Barrajón-Catalán, Junkal Garmendia

**Affiliations:** 1Instituto de Agrobiotecnología, CSIC-Gobierno Navarra, 31192 Mutilva, Spain; ariadna.fernandez@unavarra.es (A.F.-C.); beuba@alumni.unav.es (B.E.); luciacaballero21@gmail.com (L.C.); 2Centro de Investigación Biomédica en Red de Enfermedades Respiratorias (CIBERES), 28029 Madrid, Spain; mmenendez@iqfr.csic.es; 3Ikan Biotech SL, The Zebrafish Lab, Centro Europeo de Empresas e Innovación de Navarra (CEIN), 31110 Noain, Spain; roberto.diez@ikanbiotech.com; 4Instituto de Química Física Rocasolano, CSIC, 28006 Madrid, Spain; 5Laboratory of Preclinical Models and Analytical Tools, Division of Solid Tumors and Biomarkers, Center for Applied Medical Research, 31008 Pamplona, Spain; codesolorzano@unav.es; 6Centro de Investigación Biomédica en Red de Enfermedades Oncológicas (CIBERONC), 28029 Madrid, Spain; 7Servicio de Microbiología, Clínica Universidad de Navarra, 31008 Pamplona, Spain; jleiva@unav.es; 8Instituto de Investigación Sanitaria de Navarra (IdiSNA), 31008 Pamplona, Spain; 9Instituto de Investigación, Desarrollo e Innovación en Biotecnología Sanitaria de Elche (IDiBE), Instituto de Biología Molecular y Celular (IBMC), Miguel Hernández University, 03203 Elche, Spain; vmicol@umh.es (V.M.); e.barrajon@umh.es (E.B.-C.); 10Centro de Investigación Biomédica en Red de Fisiopatología de la Obesidad y Nutrición (CIBERobn), 28029 Madrid, Spain

**Keywords:** polyphenol, xenohormesis, *Haemophilus influenzae*, antimicrobial, anti-inflammatory, respiratory infection

## Abstract

Chronic obstructive pulmonary disease (COPD) is characterized by abnormal inflammation and impaired airway immunity, providing an opportunistic platform for nontypeable *Haemophilus influenzae* (NTHi) infection. In this context, therapies targeting not only overactive inflammation without significant adverse effects, but also infection are of interest. Increasing evidence suggests that polyphenols, plant secondary metabolites with anti-inflammatory and antimicrobial properties, may be protective. Here, a *Cistus salviifolius* plant extract containing quercetin, myricetin, and punicalagin was shown to reduce NTHi viability. Analysis of these polyphenols revealed that quercetin has a bactericidal effect on NTHi, does not display synergies, and that bacteria do not seem to develop resistance. Moreover, quercetin lowered NTHi airway epithelial invasion through a mechanism likely involving inhibition of Akt phosphorylation, and reduced the expression of bacterially-induced proinflammatory markers *il-8*, *cxcl-1*, *il-6*, *pde4b*, and *tnfα*. We further tested quercetin’s effect on NTHi murine pulmonary infection, showing a moderate reduction in bacterial counts and significantly reduced expression of proinflammatory genes, compared to untreated mice. Quercetin administration during NTHi infection on a zebrafish septicemia infection model system showed a bacterial clearing effect without signs of host toxicity. In conclusion, this study highlights the therapeutic potential of the xenohormetic molecule quercetin against NTHi infection.

## 1. Introduction

Chronic obstructive pulmonary disease (COPD) is an irreversible lung disease, typically caused by cigarette smoking [1]. Airflow limitation in COPD associates with chronic inflammation of the respiratory tract, a result of repeated insult by noxious components of cigarette smoke [2,3]. Respiratory inflammation is further increased in COPD patients during periods of exacerbation, which are events induced by infection in most instances [4]. The colonizing opportunistic pathogen nontypeable *Haemophilus influenzae* (NTHi) is implicated in COPD exacerbations, and frequently isolated from respiratory samples [5,6]. Repeated episodes of exacerbation and the ensuing inflammation contribute to lung damage and progressive airflow limitation, often resulting in hospitalization or death, and placing an enormous burden on health services [7,8,9].

COPD management relies on lifestyle changes, use of bronchodilators, inhaled corticosteroids (ICS), and antibiotic administration for infectious exacerbations. ICS are the mainstay of anti-inflammatory therapy, but their usefulness in COPD has been questioned due to potential side effects such as an increased risk of pneumonia [10,11,12] and corticosteroid insensitivity due to oxidative stress from cigarette smoke and chronic inflammation [13]. Long-acting bronchodilators, alone or in combination with ICS, phosphodiesterase-4 inhibitors, and lung volume reduction surgery have proven to reduce exacerbations, but a significant number of patients continue to experience acute episodes [14]. Hence, there is high demand for effective treatments to target COPD chronic inflammation, as it is hoped that reducing inflammation will lead to improved quality of life for patients and possibly reduce exacerbation frequency [15]. However, therapeutic modulation of the host immunity requires a fine-tuned balance because the same cells, molecules, and mechanisms involved in host protection can also be involved in deleterious inflammation. Moreover, excessive anti-inflammatory effects may dampen immune responses, thus facilitating infectious processes. In this context, therapies targeting not only overactive inflammation without significant adverse effects, but also infection are of particular translational significance. This is the case with macrolide antibiotics, which have anti-inflammatory effects beyond their antimicrobial activity. However, although high quality randomized controlled trials confirm that long-term azithromycin treatment decreases the risk of COPD exacerbations, careful attention needs to be paid to the potential risks of hearing decrements, cardiac toxicity, and development of microbial resistance patterns [14,15,16].

Plant metabolites acting as xenohormetic molecules are drug candidates to follow this demand, and therefore the focus of this study. Hormesis is an adaptive response in which heterotroph exposure to low doses of plant chemical compounds has a beneficial and/or adaptive effect. This response can be mediated by molecules that, when incorporated in the heterotroph diet, induce biological responses leading to pharmacological effects. Xenohormesis is this final effect as a benefit obtained by the heterotroph organism, giving us opportunities to obtain benefits from natural compounds as drugs naturally selected through evolutionary processes [17]. In particular, plant polyphenols are a large group of natural molecules with antioxidant, chelating, and anti-inflammatory properties. These molecules, which are important components of human diet, have potential benefits for cancer, cardiovascular disease, and other chronic diseases involving oxidative stress or inflammation such as rheumatoid arthritis and COPD [18,19]. In fact, high intake of catechins and solid fruits has shown a beneficial effect in COPD [20]; some plant lignans suppress the inflammatory response in cigarette smoke-stimulated airway epithelial cells and in a COPD murine model [21], and the polyphenols curcumin and quercetin attenuate cigarette smoke induced pulmonary inflammation and mouse emphysema [22,23]. Some polyphenols are also antimicrobials and may have synergistic effects, either by themselves or in combination with conventional antibiotics [24,25,26,27,28,29]. Thus, the polyphenol resveratrol has a protective role in respiratory disease, with anti-inflammatory, antioxidant, and antibacterial properties [30,31,32].

Following the interest in drugs targeting both overactive inflammation and infection, we previously evaluated the effect of azithromycin, showing that its efficacy on infection by NTHi highly relates to the minimal inhibitory concentration of the infecting strain [33], and of resveratrol, showing a protective role in NTHi infection [30]. Together, the existing evidence prompted us to screen the antibacterial effect of a panel of plant extracts with known polyphenolic composition, and to characterize the antimicrobial-immunomodulatory dual action of their specific polyphenols. *Cistus salviifolius* extract, which contains quercetin, myricetin, punicalagin, and ellagic acid as main polyphenols, reduced NTHi viability. Polyphenol analysis showed a quercetin in vitro bactericidal effect, without being prone to develop resistance. Quercetin effects were further tested on NTHi infected cultured airway epithelial and phagocytic cells, and on NTHi lung infection and septicemia in vivo model systems. The potential usefulness of modulating the host immune responses by using antimicrobial xenohormetic molecules such as quercetin is discussed.

## 2. Materials and Methods

### 2.1. Bacterial Strains, Media, Growth Conditions, and Drugs

NTHi strains were grown at 37 °C, 5% CO_2_ on PVX agar (Biomérieux), or on *Haemophilus* Test Medium (HTM) Base agar (Oxoid) supplemented with 10 μg/mL hemin and 10 μg/mL nicotinamide adenine dinucleotide (NAD) (Sigma-Aldrich), referred to as sHTM agar. NTHi liquid cultures were grown in brain heart infusion (BHI) (Oxoid) supplemented with 10 μg/mL hemin and 10 μg/mL NAD, referred to as sBHI. NTHi375 is a clinical isolate from the middle ear of a pediatric patient with otitis media [34]. When necessary, bacterial viability upon host cell infection conditions was tested. To do so, phosphate-buffered saline (PBS)-normalized bacterial suspensions (OD_600_ = 1~10^9^ c.f.u./mL) were prepared by using NTHi grown on PVX agar, 100 μL aliquots were incubated in 1 mL Earle’s Balanced Salt Solution (EBSS, Gibco) medium in the absence/presence of selected polyphenol concentrations for 2 h at 37 °C, serially diluted, and plated on sHTM agar for c.f.u./mL determination. When necessary, heat killed (HK) bacteria were used. For this purpose, a bacterial suspension was recovered from a freshly grown PVX agar plate with 1 mL PBS, adjusted to OD_60_  = 1 and incubated at 80 °C for 30 min. Ampicillin (Amp, Sigma-Aldrich) and azithromycin dihydrate (Azm, Zytromax) 10 mg/mL stock solutions were prepared in distilled H_2_O (dH_2_O) and filtered. Four plant extracts rich in polyphenols were used: *Cistus salviifolius* extract (45% *w*/*w* polyphenols) was obtained as previously described [35]; *Punica granatum* extract (20% *w*/*w* punicalagins) was provided by Nutracitrus S.L. (Elche, Spain); *Hibiscus sabdariffa* (6% *w*/*w* polyphenols) and *Lippia citriodora* (30% *w*/*w* phenylpropanoids) extracts were provided by Nutrafur S.L. (Alcantarilla, Spain). These extracts were used by preparing 5 mg/mL stock solutions, as follows: *P. granatum*, *L. citriodora* and *H. sabdariffa* extracts were dissolved in dH_2_O and filtered; *C. salviifolius* extract was dissolved in dH_2_O:DMSO (9:1, *v*/*v*) and sonicated in an ultrasonic bath for 1 min. Plant extract stocks were freshly prepared prior use. Quercetin (purity ≥95%), myricetin (purity ≥96%) (Sigma-Aldrich) and punicalagin (A+B mixture, product #80524 PhytoLab) were used by preparing 10 mg/mL stock solutions in methanol (for “only bacteria” assays) or DMSO (for host cell infection and in vivo assays), and preserved at −20 °C until use. Antibiotic, plant extract, and pure polyphenol working concentrations are specified in each type of assay (see below).

### 2.2. Determination of Plant Extract and Pure Polyphenol Antimicrobial Effects

A broth microdilution assay was developed to determine minimal inhibitory concentrations (MIC). To do so, 500 μg/mL working solutions were prepared in sBHI for *P. granatum*, *L. citriodora* and *H. sabdariffa* extracts, and for quercetin, myricetin, and punicalagin; differently, a *C. salviifolius* 500 μg/mL working solution was prepared in PBS. Then, 120 µL aliquots were transferred to individual wells in column 1 of 96-well microtiter plates. A vehicle solution control consisting of a volume of dH_2_O, dH_2_O:DMSO (9:1, *v*/*v*) or methanol, equivalent to that used for the highest plant extract/polyphenol concentration tested, was performed in parallel. From columns 2 to 12, 80 μL aliquots of sBHI were transferred to individual wells when performing *P. granatum*, *L. citriodora*, *H. sabdariffa*, quercetin, myricetin, and punicalagin assays, and 80 μL aliquots of PBS were transferred to individual wells when performing *C. salviifolius* assays. Next, 40 μL serial dilutions were made from columns 1 to 12, by discarding 40 μL from column 12, to have 80 μL per well in all cases. When the MIC values ranged between 500 and 167 μg/mL, a range of intermediate concentrations was manually prepared. A suspension of PVX agar freshly grown bacteria was generated in PBS (to be used for testing *P. granatum*, *L. citriodora*, *H. sabdariffa* extracts, and quercetin, myricetin and punicalagin polyphenols) or in sBHI (to be used for testing *C. salviifolius* extract), adjusted to OD_600_ = 1, serially diluted in PBS or sBHI, respectively to 10^−6^. Then, 20 μL bacterial aliquots were transferred to individual wells. Blank controls (PBS or sBHI without bacteria) were used per each tested sample type and concentration. Plates were incubated for 24 h at 37 °C, 5% CO_2_, without agitation. Next, 25 μL aliquots of INT-formazan (Sigma-Aldrich, 1 mg/mL solution prepared in dH_2_O), were transferred to individual wells. Viable bacteria render a red precipitate after 1 h incubation at 37 °C, 5% CO_2_, without agitation. Bacterial survival was quantified by measuring OD_570_; absorbance values were corrected to their matching blanks values. Percentage (%) of bacterial survival was calculated according to each corresponding vehicle solution control. At least three independent assays were performed in duplicate (n ≥ 6).

### 2.3. Determination of Antimicrobial Synergic Effects

Serial dilutions of each molecule to be tested were prepared in sBHI and combined at different proportions following the checkerboard method [36,37]. Briefly, after generating each polyphenol-polyphenol or polyphenol-antibiotic concentration matrix, each individual well of 96-well microtiter plates contained 80 µL final volume. A suspension of PVX agar freshly grown bacteria was generated in PBS, adjusted to OD_600_ = 1, serially diluted in PBS to 10^−6^, and 20 µL aliquots were transferred to individual wells. Plates were incubated for 24 h at 37 °C, 5% CO_2_, without agitation. Bacterial viability was determined by using the INT-formazan reagent as readout (see above). It was also used to calculate the fractional inhibitory concentration (FIC) index of each polyphenol-polyphenol or polyphenol-antibiotic combination, and to determine the existence of synergy (ΣFIC ≤ 0.5), additive (0.5 > ΣFIC ≤ 1), indifferent (1 > ΣFIC < 2) or antagonic (ΣFIC ≥ 2) effects. At least 3 independent assays were performed (n ≥ 3).

### 2.4. Polyphenol Susceptibility Assays

To assess if the polyphenol antimicrobial effect is bactericidal or bacteriostatic, quercetin, myricetin, and punicalagin solutions with concentrations equivalent to each respective MIC were prepared in sBHI, and 80 µL aliquots were transferred to individual wells in 96-well microtiter plates. As a bactericidal control, Amp 4 μg/mL was prepared in sBHI, and 80 µL aliquots were transferred to individual wells in 96-well microtiter plates. A vehicle solution control consisting of a volume of methanol equivalent to that used for each polyphenol concentration tested was performed in parallel. A suspension of PVX agar freshly grown bacteria was generated in PBS, adjusted to OD_600_ = 1, serially diluted in PBS to 10^−6^, and 20 µL aliquots were transferred to individual wells in 96-well microtiter plates. Plates were incubated for 24 h at 37 °C, 5% CO_2_, without agitation. Next, 20 μL aliquots from each well were transferred to their matching wells on a new microplate, where 80 μL sBHI aliquots had been previously added. Plates were incubated as before, and bacterial viability determined with the INT-formazan reagent in all cases. Wells with viable bacteria were used for serial dilution, plating on sHTM agar, and c.f.u./mL determination. Three independent assays were performed in quadruplicate (n = 12).

### 2.5. Serial Passage Experiment with Polyphenol

This assay was adapted from [38]. Briefly, quercetin, myricetin, and punicalagin stock solutions prepared in methanol were diluted in sBHI to quercetin, 100 μg/mL; myricetin, 256 μg/mL; punicalagin, 205 μg/mL. Such concentrations are subinhibitory for each tested polyphenol. 80 μL of each polyphenol concentration were transferred to individual wells in 96-well microtiter plates. A vehicle solution control consisting of a volume of methanol equivalent to that used for each polyphenol concentration tested was performed in parallel. A suspension of PVX agar freshly grown bacteria was generated in PBS, adjusted to OD_600_ = 1, serially diluted in PBS to 10^−6^, and 20 µL aliquots were transferred to individual wells. Blank controls (PBS) were used per each tested sample type. Plates were incubated for 24 h at 37 °C, 5% CO_2_, without agitation. Cultures were then passaged (20 μL in 80 μL fresh sBHI with polyphenol or vehicle solution) every day for 15 days. At each time point throughout the cycling, each well absorbance (OD_600_ at 24 h–OD_600_ at 0 h) was measured. Four replicates per condition were made in two independent experiments (n = 8).

### 2.6. Infection of Tissue Cultured Cells

Carcinomic human alveolar basal epithelial cells (A549, ATTC CCL-185) were maintained as described previously [39], seeded to 1.5 × 10^5^ cells per well (24-well plates) for 32 h, and then serum starved for 16 h before infection. MH-S murine alveolar macrophages (ATCC CRL-2019) were maintained as described previously [40] and seeded to 5 × 10^5^ cells per well in 24-well plates 16 h before infection. For infection, PBS-normalized bacterial suspensions (OD_600_ = 1) were prepared by using NTHi grown on PVX agar. Adhesion, invasion (for A549 cells), and phagocytosis (for MH-S cells) assays were performed as previously described [39,40,41]. When indicated, cells were pretreated for 4 h with quercetin or vehicle solution (DMSO) (in EBSS for A549 cells; in RPMI 1640 with 10 mM HEPES and 10% heat-inactivated FCS for MH-S cells). The polyphenol-containing medium was replaced by polyphenol-free medium prior infection. Alternatively, cells were infected, and quercetin or DMSO was added during the gentamicin (Gm) incubation period. Quercetin did not induce cytotoxicity, determined by measuring the release of lactate dehydrogenase and microscopy (data not shown). Controls (indicated as V in the Figures) were performed by using DMSO volumes corresponding to that of the highest drug concentration tested in each assay. After bacterial infection, wells were washed and cells were lysed as previously described [39]. Lysates were serially diluted in PBS and plated on sHTM agar for bacterial counts. Results are expressed as c.f.u./well. Experiments were performed in triplicate and on at least three independent occasions (n ≥ 9).

### 2.7. RNA Extraction and Real-Time Quantitative PCR

A549 cells were pretreated with quercetin/vehicle solution (DMSO) for 4 h, and incubated with HK NTHi (equivalent in numbers to that of the infecting dose, drug exposure was maintained during bacterial-cell contact) for 2 h in EBSS, as described previously [30]. MH-S cells were pretreated with quercetin/DMSO for 4 h and incubated with HK NTHi for 1 h in RPMI 1640 with 10 mM HEPES and 10% heat-inactivated FCS. Uninfected groups were included as controls. Next, total RNA was isolated from cells using a Nucleospin RNAII kit (Macherey-Nagel, Düren, Germany) as recommended by the manufacturer, and including an on column DNase treatment step. When indicated, total RNA was isolated from mouse lungs using TRIzol reagent (Invitrogen, Carlsbad, CA, USA). Total RNA quality was evaluated using RNA 6000 Nano LabChips (Agilent 2100 Bioanalyzer, Santa Clara, CA, USA). All samples had intact 18S and 28S ribosomal RNA bands with RNA integrity numbers (RIN) between 7.5 and 10. Reverse transcription was performed using 1 μg of RNA by PrimeScript RT Reagent kit (Takara, Shiga, Japan). To amplify human *il-6*, *il-8*, *cxcl-1*, *pde4b*, *gapdh*, and mouse *kc*, *tnfα*, *pde4b*, *gapdh* genes, 1:10 diluted cDNA were used as template (including endogenous control). In all cases, 20 μL reaction mixtures containing 1X SYBR Premix Ex Taq II (Tli RNaseH Plus) (Takara) and the adequate primer mix were used. Fluorescence data were analyzed with AriaMx Real-Time PCR System (Agilent Technologies, Santa Clara, CA, USA). The comparative threshold cycle (Ct) method was used to obtain relative quantities of mRNA that were normalized using human or mouse *gapdh* as an endogenous control. Intron-spanning primers were designed with Primer-BLAST software (NCBI) (Appendix A). All measures were performed in triplicate and at least three times (n ≥ 9).

### 2.8. Western Blotting

A549 cells seeded as above indicated were pretreated with quercetin 60 μg/mL or vehicle solution (DMSO) for 4 h in EBSS, and quercetin was removed prior infection. For infection, PBS-normalized bacterial suspensions (OD_600_ = 1) were prepared by using NTHi strains grown on PVX agar, and 100 μL aliquots were transferred to each well. Cells were infected for 0 (uninfected), 10, 20, 30, 45 and 60 min. Afterwards, the wells were washed 3 times with cold PBS, lysed with 50 μL of lysis buffer (62.5 mmol Tris-HCl pH 6.8, 2% *w*/*v* SDS, 10% glycerol, 50 mmol DTT, 0.01% *w*/*v* bromophenol blue) and scraped on ice. Samples were sonicated, boiled, and cooled on ice before 10% SDS-PAGE and western blotting. Akt phosphorylation was detected with primary rabbit anti-phosphoSer473 Akt antibody (Cell Signaling Technology, Beverly, MA, USA) diluted 1:1000. Total Akt, used as a loading control, was detected with primary rabbit anti-Akt antibody (Cell Signaling Technology) diluted 1:1000. A secondary goat anti-rabbit antibody conjugated to horseradish peroxidase (Thermo Scientific, Rockford, IL, USA) diluted 1:10000 was used. ECL AdvanceTM Western Blotting Detection Kit (GE HealthCare, Boston, MA, USA) was used for detection. Western blots were performed at least three times by using independently generated cell extracts (n = 3). Images corresponding to a representative experiment are shown in the results section.

### 2.9. NTHi Mouse Lung Infection

A mouse model of NTHi lung infection was used as previously described [42]. CD1 female mice (18–20 g) aged 4–5 weeks purchased from Charles River Laboratories (France) were housed under pathogen-free conditions at the Institute of Agrobiotechnology facilities (registration number ES/31-2016-000002-CR-SU-US). Animal handling and procedures were in accordance with the current European and National (RD 53/2013) legislation (Protocol PI 007/19). Emphysema was induced by intratracheal administration of porcine pancreatic elastase (EPC, Elastin Products Company, Owensville, MI, USA). To do so, 10 mg containing 1350 elastase units (U) were resuspended in 10 mL physiological serum to generate a stock solution (1 mg/mL, i.e., 135 U/mL). To induce emphysema, one 90 μL dose containing 6 elastase U/mouse was administered 21 days before infection. Quercetin treatment was performed at 60 mg/kg of body weight in 0.1 mL PBS-DMSO (1:1) and administered by oroesophageal gavage (Popper&Sons Inc., Philadelphia, PA, USA). Administrations were performed daily during 8 days before infection (1 h before infection on day 8), and at 6, 12 and 23 h post-infection (hpi). Quercetin administration at 6 hpi was performed in mice euthanized at 12 hpi; quercetin administration at 6, 12 and 23 hpi was performed in mice euthanized at 24 hpi. NTHi strain 375 was used for lung infection. Mice were randomly divided into infected and control uninfected groups, and into quercetin-treated and vehicle solution (PBS-DMSO, 1:1)-administered mice. Animals were euthanized at 12 or 24 hpi (n ≥ 6 per group). For NTHi intranasal infection, 20 μL of an exponentially grown (OD_600_ = 0.3) bacterial suspension containing ~5 × 10^9^ c.f.u./mL (~1 × 10^8^ c.f.u./mouse) was placed at the entrance of the nostrils until complete inhalation by the mouse, previously anesthetized (ketamine-xylazine, 3:1). At the indicated time points, mice were euthanized and lungs were aseptically removed. The left lung was individually weighed in sterile bags (Stomacher80, Seward Medical Ltd., Rhymney, UK) and homogenized 1:10 (*w*/*v*) in PBS. Each homogenate was serially 10-fold diluted in PBS and plated in triplicate on sHTM agar to determine the number of viable bacteria. The right lung was homogenized with TRIzol reagent for subsequent RNA extraction as indicated above.

### 2.10. NTHi Adult Zebrafish Infection

Animal experiments conducted at Ikan Biotech (https://www.ikanbiotech.com) animal housing facility were performed as previously described [30], according to the approval of the Universidad de Navarra (UNAV) Ethics Committee for Animal Experimentation (Protocol 107-19). Quercetin toxicity in zebrafish embryos was tested by performing the OECD TG236 “Fish embryo acute toxicity (FET) test”. Once lack of quercetin toxicity in zebrafish was confirmed (data not shown), adult zebrafish were randomly divided into 2 infected (administered quercetin or vehicle solution) and 2 uninfected (administered quercetin or vehicle solution) groups (n ≥ 6 per group). Infected groups were injected with 10 μL of a NTHi375 exponentially grown (OD_600_ = 0.3) suspension containing ~10^10^ c.f.u./mL (~10^8^ c.f.u./zebrafish), prepared in perfusion solution (Grifols, Spain). At 29 and 53 hpi, an infected and an uninfected group were intraperitoneally administered quercetin at a dose of 0.3 mg/g of body weight in 10 μL of perfusion solution-DMSO (1:1); the other groups were administered perfusion solution-DMSO (1:1). Survival rate for each group was monitored three times per day for 5 days after infection.

### 2.11. Statistical Analysis

In all cases, *p* < 0.05 value was considered statistically significant. Analyses were performed using Prism software, version 7 for Mac (GraphPad Software) statistical package, and are detailed in each Figure legend.

## 3. Results

### 3.1. An Extract of Cistus salviifolius Rich in Polyphenols Has an Antimicrobial Effect on H. influenzae

*Cistus salviifolius* and *Punica granatum* extracts, containing ellagitannins and flavonoids, were selected for this study due to their previously seen antibacterial effects [29,43]. *Hibiscus sabdariffa* and *Lippia citriodora* extracts were selected based on their antifungal activity and their efficacy against some bacteria [35]. We first tested the susceptibility of the clinical strain NTHi375 to these four plant extracts rich in polyphenols and observed a significant dose-dependent reduction of bacterial viability after incubation with the *C. salviifolius* extract (Figure 1A). This observation was also made for the reference strain RdKW20, which also showed lower viability at the highest *P. granatum* concentrations tested (Appendix A). These observations prompted us to analyze the effect of three polyphenols, quercetin, myricetin, and punicalagin, the most representative aglycones of the large variety of glycosylated polyphenols present in the *C. salviifolius* extract [35]. The three molecules reduced NTHi bacterial viability in a dose-dependent manner, quercetin being the one with the lowest MIC observed (Figure 1B and Appendix A). Genomic heterogeneity is a known feature for NTHi [44,45,46], which may lead to variable polyphenol susceptibility among strains, as is shown for other antimicrobials [33]. We evaluated the effect of quercetin, myricetin, and punicalagin on three NTHi clinical strains isolated from COPD sputum samples and belonging to different clonal types [45]; their viability decreased when exposed to polyphenols, with some MIC variations among them (Appendix A).

These results support that the extract of *C. salviifolius* and, in particular, three of its containing polyphenols, quercetin, myricetin, and punicalagin, reduce NTHi viability in a dose-dependent manner.

### 3.2. Antimicrobial Effects of Quercetin, Myricetin, and Punicalagin against NTHi

Next, we evaluated if polyphenol effects on NTHi viability are bactericidal or bacteriostatic. NTHi375 cells were incubated with quercetin, myricetin or punicalagin (MIC concentrations, based on observations shown in Figure 1B) for 24 h, bacterial cultures were then diluted into new microplate wells without polyphenol for 24 additional hours, and the number of viable cells was monitored by serial dilution and plating at the indicated time intervals (24 and 48 h). After 24 h, quercetin, myricetin, and punicalagin-treated cultures showed a lower number of viable cells than the vehicle solution control ones. At 48 h, only quercetin-treated cultures showed a lower number of viable cells than the control ones. In contrast, myricetin and punicalagin-treated cultures showed growth numbers comparable to those of the control ones. These results suggested quercetin to be bactericidal, and myricetin and punicalagin to be bacteriostatic on NTHi375 (Figure 1C). Similar results were observed for RdKW20 where, besides quercetin, punicalagin seemed to be bactericidal as well (Appendix A).

When quercetin, myricetin, and punicalagin were combined at different proportion and tested against NTHi375 by the checkerboard method, antagonism or synergy effects were not detected. The same result was obtained when the three polyphenols were independently combined with the antibiotics azithromycin (Azm) or ampicillin (Amp). Results in Table 1 show the MIC ± SD for each compound separately, the MIC ± SD of the combinations tested, calculated ΣFIC ± SD, and their interpretation.

We next assessed the ability of NTHi to become resistant to quercetin, myricetin, or punicalagin through serial independent passage of strain NTHi375 in sBHI broth containing subinhibitory concentrations (<MIC) of each polyphenol, and using Amp and vehicle solution as controls. After 15 consecutive overnight passages, no growth was detected (Figure 1D).

In summary, quercetin presented a bactericidal activity and, under the conditions tested, it did not seem to induce resistance of NTHi375. Based on these observations, quercetin was shortlisted to next assess its modulatory effect on the NTHi-host airway interplay. NTHi375 was used for this purpose, given that it has been previously used in host-pathogen interaction studies [42,47,48,49].

### 3.3. Quercetin Modulates NTHi Infection of Cultured Airway Epithelial Cells

Quercetin is known to modulate eukaryotic cell signaling by inhibiting Akt phosphorylation [50] and phosphodiesterase 4 (PDE4) [51]. Akt phosphorylation has been reported to facilitate- and increased intracellular cAMP to reduce NTHi epithelial invasion, respectively [39,41,52]. Based on previously known quercetin modulatory effects, we hypothesized that host cell treatment with this polyphenol may reduce NTHi epithelial infection. First, quercetin effect on bacterial viability was titrated in EBSS medium aiming to mimic cultured epithelial cell infection conditions (see Materials and Methods section). A significant decrease on NTHi375 bacterial viability was observed at quercetin concentrations as low as 3 μg/mL (Figure 2A). Such quercetin concentrations did not cause a cytotoxic effect on A549 cells upon 6 h cell exposure (data not shown). Based on these observations, cells were pretreated with EBSS containing quercetin for 4 h, which was replaced by EBSS medium without drug prior infection. Exposure to increasing doses of quercetin reduced NTHi375 epithelial adhesion and invasion rates in a dose-dependent manner. Adhesion data showed a clear but non-significant trend, and invasion data showed a significant reduction upon cell pretreatment with quercetin 60 μg/mL (Figure 2B,C). We also monitored quercetin effect on Akt phosphorylation during NTHi infection. Quercetin was shown to inhibit Akt phosphorylation under the conditions tested (Figure 2E), which may at least partially contribute explaining the observed diminished bacterial cell invasion.

Quercetin eukaryotic cell permeation may occur by a passive diffusion mechanism [53]. Given that antimicrobial penetrating epithelial cells can reduce intracellular NTHi viability [33], we asked if quercetin could reduce the viability of internalized bacteria. To do so, A549 cell infection with NTHi375 was performed in culture medium without drug supplementation, followed by subsequent incubation in fresh medium supplemented with gentamicin to kill extracellular bacteria, and with vehicle solution or quercetin, 30 or 60 μg/mL. Under these conditions, quercetin did not reduce the number of intracellular bacteria (Figure 2D).

Together, these results show that quercetin sub-inhibitory concentrations modulate NTHi375 airway epithelial cell infection, and quercetin-driven inactivation of Akt may play a role in such interference. Under the conditions tested, quercetin does not modify already internalized bacterial numbers.

### 3.4. Quercetin Lowers Proinflammatory Gene Expression by Epithelial Cells Infected with H. influenzae

Quercetin has shown to reduce proinflammatory cytokine production in *Vibrio cholerae* infected intestinal epithelial cells [54], and in *Streptococcus suis* infected macrophages [55]. To determine if quercetin modulates the expression of genes encoding proinflammatory cytokines in human airway epithelial cells followed by NTHi infection, A549 cells were pretreated with quercetin. To maintain the polyphenol at the onset of the infectious inflammatory stimulus without jeopardizing bacterial viability, we assessed gene expression by cells incubated with heat killed (HK) bacteria. Under these conditions, expression of the *il-8*, *cxcl-1* and *il-6* genes was stimulated, compared to control uninfected cells, and such stimulation was significantly reduced when cells were treated with quercetin in a dose-dependent manner (Figure 3A–C).

Conversely, therapeutic inhibition of PDE4 has a variety of known anti-inflammatory effects beneficial in the treatment of COPD [56]. Given that quercetin is a naturally occurring selective PDE4 inhibitor [51], and that NTHi airway epithelial infection stimulates differential expression for several PDEs including PDE4B (the major PDE isoform expressed in lung) [52], the question arises as to if quercetin’s anti-inflammatory effect on NTHi-infected cells could also be mediated by modulating *pde4b* gene expression. As shown in Figure 3D, *pde4b* gene expression was increased in cells incubated with HK bacteria for 2 h, when compared to control uninfected cells, and quercetin treatment significantly lowered *pde4b* gene expression in a dose-dependent manner.

In conclusion, these results show quercetin likely anti-inflammatory effects when maintained at the onset of the infectious inflammatory stimulus, as a means of lowering *il-8*, *cxcl-1*, *il-6*, and *pde4b* gene expression.

### 3.5. Antimicrobial and Anti-inflammatory Effects of Quercetin Administration on Mouse Lung Infection with NTHi

Next, we sought to determine the effect of quercetin oral administration in vivo, by NTHi respiratory infection of mice with previously induced lung emphysema. We used a regimen of oral quercetin (60 mg/kg) consisting of daily administrations during 8 days prior to infection and three administrations at 6, 12 and 23 hpi (Figure 4A). We observed a moderate reduction in NTHi counts following treatment with quercetin, with a non-statistically significant but reproducible trend to be lower in quercetin-treated than in control untreated animals (Figure 4B). We asked if such moderate reduction in NTHi counts following quercetin administration may relate to impaired phagocytosis by alveolar macrophages. However, MH-S cell exposure to quercetin did not modify NTHi uptake, despite showing a significant anti-inflammatory effect by the means of reducing the expression of the *tnfα* gene in cells incubated with HK bacteria (Appendix A).

Next, we analyzed NTHi-induced expression of *kc* and *tnfα* proinflammatory mediators in mice lung tissue euthanized at 12 hpi, which rendered higher numbers than those obtained for control uninfected animals. In turn, treatment with quercetin decreased NTHi-induced expression of *kc* and *tnfα* in infected mice lung tissue. The *pde4b* gene expression was also increased in lung tissue of NTHi-infected mice, compared to control non-infected animals, and quercetin treatment decreased NTHi-induced expression of *pde4b* in infected lung tissue. No differences in terms of *kc*, *tnfα,* or *pde4b* gene expression were observed in quercetin treated and control untreated non-infected animals (Figure 4C).

In summary, these results show that quercetin 60 mg/kg moderately reduces bacterial load, and lowers the expression of whole-lung inflammatory markers.

### 3.6. Quercetin Antimicrobial Protective Effect on Zebrafish Systemic Infection with NTHi

Lastly, a quercetin acute toxicity assay was performed in zebrafish embryos following the fish embryo acute toxicity (FET) test. Results showed that the highest quercetin non-toxic dose was 7 μg/mL (data not shown). We then performed a quercetin assay in adult zebrafish, following a previously established sepsis model system by intraperitoneal NTHi infection [30]. We assessed quercetin antimicrobial effect on NTHi375 infected zebrafish by using a therapeutic regimen of intraperitoneal quercetin (0.3 mg/g) consisting of two administrations at 29 and 53 hpi. Survival rate for quercetin-treated and control untreated groups was monitored up to 5 days post-infection. Mortality rate in quercetin-treated infected zebrafish was significantly lower than in infected animals receiving vehicle solution (Figure 5). This model system rendered significant quercetin-mediated increased survival upon zebrafish NTHi infection.

## 4. Discussion

Chronic inflammation and NTHi infectious exacerbations are two major pathophysiological traits of COPD progression. Therapeutics targeting both infection and overactive inflammation at the COPD airway are of interest, and even more so when clinical evidence supports corticosteroid therapeutic failure not only associated with increased risk for pneumonia, but also to oxidative stress- and *H. influenzae*-mediated insensitivity [10,11,12,13,57]. Benefits of drugs with dual antimicrobial/anti-inflammatory properties have been mostly exploited for the macrolide antibiotic azithromycin, whose long-term low-dose use to prevent and/or manage COPD exacerbations may be beneficial, but could also increase the rate of macrolide resistance and cause adverse effects [58].

In the present study, our search of plant xenohormetic molecules effective against NTHi led us to focus on quercetin, a flavonoid ubiquitously present in vegetables, fruits, tea, and wine, claimed to exert beneficial effects linked to being one of the most prominent dietary antioxidants, with high accumulation observed in rat lung tissue [59]. Protective effects of quercetin in lung lesions have been reported on lipopolysaccharide-induced acute lung injury and elastase-induced emphysema murine model systems [22,60,61]. Moreover, quercetin seems to restore corticosteroid sensitivity in cells from COPD patients by activation of the adenosine monophosphate-activated protein kinase (AMPK) [62], and a community clinical trial showed a reduction of upper respiratory tract infection total sick days and severity in middle age and older subjects ingesting 1000 mg quercetin/day for 12 weeks [63]. Quercetin antimicrobial effect has been previously reported [64,65,66] but, to our knowledge, this is the first study assessing its antimicrobial activity against *H. influenzae*.

From the antimicrobial perspective, quercetin showed a bactericidal effect on *H. influenzae*, and growth was not detected on a serial passage assay, suggesting that this pathogen does not develop resistance. However, we did not observe synergism, polyphenol-polyphenol or polyphenol-antibiotic, different to previously reported synergistic activities against *Pseudomonas aeruginosa*, *Acinetobacter baumannii* or *Staphylococcus aureus* [28,64]. From the immunomodulatory perspective, quercetin reduced the expression of key proinflammatory markers in both epithelial cells and alveolar macrophages upon NTHi bacterial infection. Such an anti-inflammatory effect could also be seen in vivo upon pulmonary infection of emphysema mice, although the quercetin impact on NTHi bacterial counts was only moderate in this experimental setting. In vivo treatment of mouse infection with quercetin, by using regimens ranging from 25 to 150 mg/kg, has reported fairly variable results, from *Streptococcus suis* virulence reduction and suppression of lethal shock upon *Salmonella typhimurium* infection [55,67], to a moderate decrease in *Helicobacter pylori* counts [68], no effect on bacterial counts in a mouse model of periodontitis by *Aggregatibacter actinomycetemcomitants* [69], or even increased *Chlamydia pneumoniae* counts in murine lungs [70]. Here, quercetin administration showed a non-significant trend towards reducing NTHi bacterial counts in mouse lung homogeneates, and a clearer protective effect on the zebrafish sepsis model system. Our previous study showed that administration of resveratrol reduced bacterial counts in both mice and zebrafish model systems, and it did not modify expression of the *pde4b* gene [30], whose inactivation is a known anti-inflammatory procedure in COPD therapeutics, and could indeed be observed upon quercetin treatment in both cultured epithelial cells and murine lung homogenates upon NTHi infection.

Thus, the usefulness of modulating COPD immune responses by using antimicrobial xenohormetic molecules highlights that the therapeutic benefit of molecules with dual antimicrobial and immunomodulatory properties is likely to require the fine-tuned balance of both effects. A delicate balance between immunity and inflammation is therefore required for making it possible to fight pathogens effectively while limiting inflammation that might be damaging to the host. We hereby put forward the notion that such bilateral (bacterial burden and host immunomodulation) analysis should always be performed when exploring anti-inflammatory molecules to achieve unbiased information. In fact, the non-antimicrobial corticosteroid dexamethasone attenuates inflammation after NTHi challenge, but also increases bacterial burden [71], and the effect of the polyphenolic immunomodulator curcumin on NTHi bacterial loads is unknown [72,73] and should probably be revisited.

Finally, an investigation of plant extract or pure polyphenol administration route, body distribution, bioavailability, pharmacokinetics, and synergistic effects with conventional antimicrobials in vivo is necessary. When orally administering the *C. salviifolius* extract, punicalagins will be poorly bioavailable, and quercetin, upon absorption, will be subjected to different types of metabolism with quercetin-3-O-*β*-D-glucuronide and quercetin aglycone as the main pharmacologically active metabolites in plasma, in a similar fashion to pure quercetin oral administration [74,75].

In conclusion, we present here the therapeutic potential of polyphenol containing plant extracts, and of quercetin in particular, against NTHi infection. Further work will help to better define polyphenol lead candidates of potential use in counteracting chronic airway diseases undergoing infectious exacerbations.

## Figures and Tables

**Figure 1 biomolecules-09-00891-f001:**
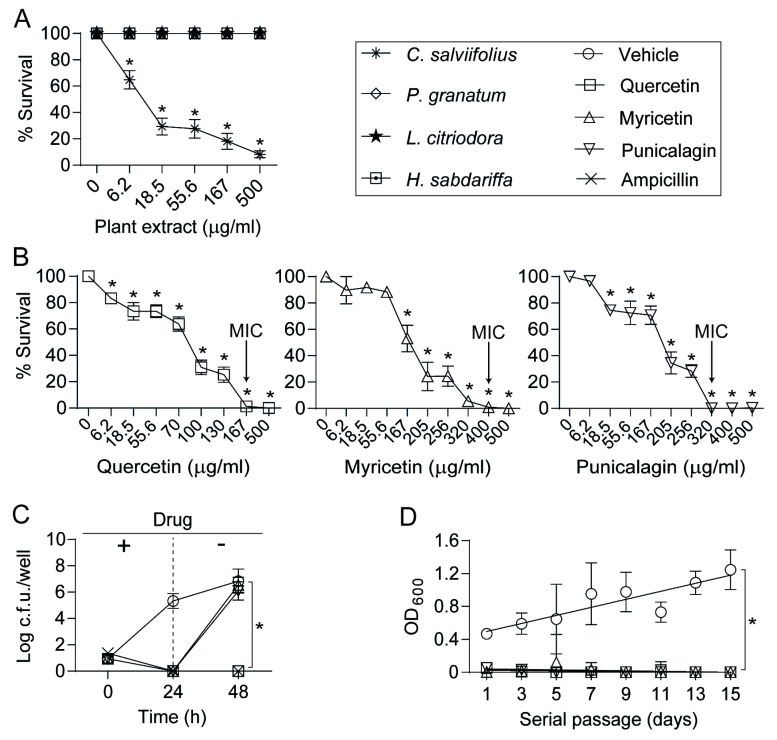
Antimicrobial effect of plant extracts and pure polyphenols on *H. influenzae*. (**A**) NTHi375 strain is susceptible to *Cistus salviifolius* extract in a dose-dependent manner (* *p* < 0.001). Results are shown as the survival percentage (mean ± SEM); (**B**) Polyphenols present in the *C. salviifolius* extract, quercetin (* *p* < 0.001), myricetin (* *p* < 0.0001) and punicalagin (* *p* < 0.0001), reduced NTHi375 survival in a dose-dependent manner. Survival percentage (mean ± SEM) is shown; (**C**) Quercetin has a bactericidal effect, and myricetin and punicalagin have a bacteriostatic effect on NTHi375, when comparing bacterial counts (log c.f.u./well, mean ± SD) after incubation with- and without polyphenol inhibitory concentrations. At 24 h, only bacteria incubated with vehicle solution rendered counts, when compared with ampicillin (Amp)-treated cultures (* *p* < 0.0001). After polyphenol replacement by sBHI, 48 h bacterial cultures previously incubated with punicalagin or myricetin rendered significant counts, when compared with quercetin and Amp treated cultures (* *p* < 0.0001); (**D**) NTHi375 did not grow after 15 daily serial passages in the presence of polyphenol (quercetin, myricetin or punicalagin) subinhibitory concentrations. Data are shown as OD_600_ (mean ± SD) in every passage (* *p* < 0.0005). Statistical comparisons of the means were performed with two-way ANOVA (**A** and **C**) or one-way ANOVA (**B** and **D**), and Dunnett’s multiple comparisons test.

**Figure 2 biomolecules-09-00891-f002:**
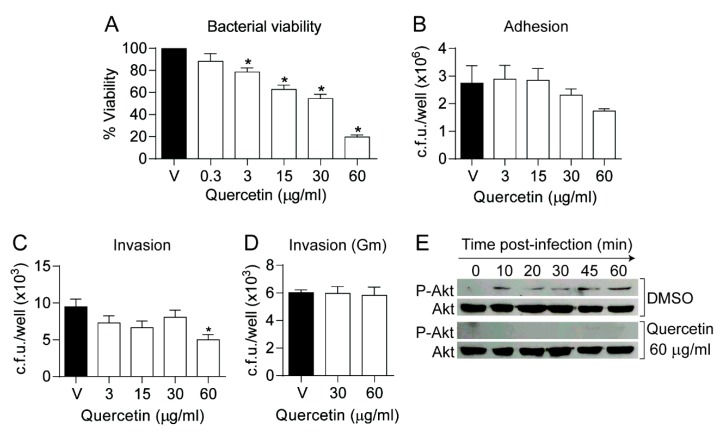
Quercetin modulates *H. influenzae* airway epithelial cell infection. (**A**) NTHi375 viability was tested by simulating host cell infection conditions (2 h at 37 °C in EBSS medium, in the absence (V)/presence of quercetin). Quercetin reduced (* *p* < 0.0001) bacterial viability (mean ± SEM) in a dose-dependent manner. Based on these results, cell infection assays shown in panels **B**, **C** and **E** were performed by cell pretreatment with quercetin for 4 h, and drug removal prior infection. Controls (V): cells did not receive quercetin, but did receive vehicle solution, i.e., DMSO; (**B**) Adhesion assays did not render significant differences, despite showing a reproducible trend toward lower adherent bacterial counts; (**C**) A significant reduction of NTHi375 epithelial invasion was observed in cells pretreated with quercetin 60 µg/mL (* *p* < 0.01); (**D**) Quercetin effect on intracellular NTHi375. A549 cells were infected and quercetin 30 or 60 μg/mL was added during cell incubation with gentamicin (Gm). Quercetin did not reduce the number of intracellular bacteria. Results are shown as c.f.u./well (mean ± SEM). Statistical comparisons of the means were performed with one-way ANOVA and Dunnett’s multiple comparisons test; (**E**) Quercetin impairs Akt phosphorylation during NTHi infection. Quercetin-treated cells showed no Akt activation upon infection, opposite to DMSO-treated cells showing time-dependent Akt phosphorylation. A representative western-blot showing P-Akt (activated form) and total Akt (loading control) is shown.

**Figure 3 biomolecules-09-00891-f003:**
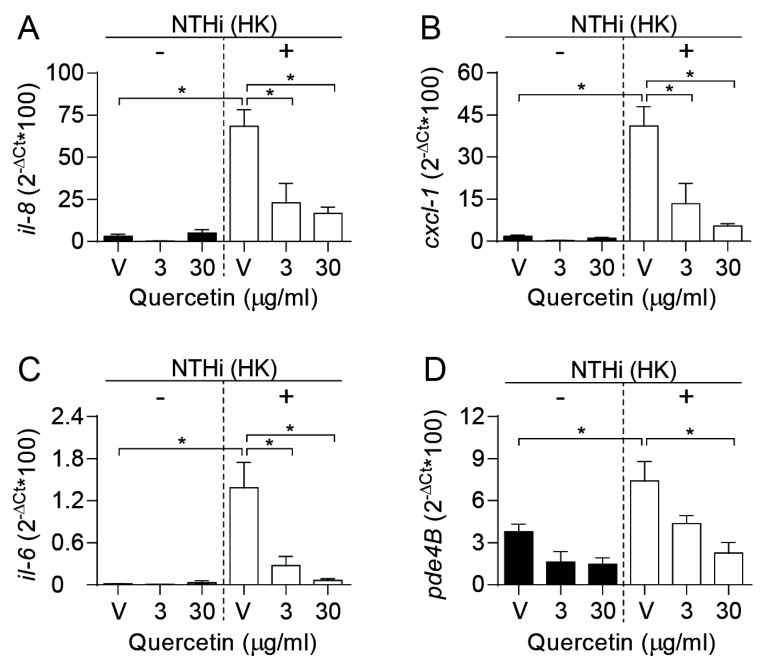
Quercetin has an anti-inflammatory effect on NTHi-infected airway epithelial cells. A549 cells were pretreated with quercetin or DMSO (vehicle solution) for 4 h; next, heat killed (HK) bacteria were used as an inflammatory stimulus (NTHi HK) for 2 additional hours. Black bars, non-infected; white bars, infected cells. V refers to cells exposed to vehicle solution. To monitor gene expression, relative quantity of human *il-8* (**A**), *cxcl-1* (**B**), *il-6* (**C**) and *pde4b* (**D**) mRNA was measured by qRT-PCR. In all cases, gene expression was significantly increased upon infection of control untreated cells (* *p* < 0.0001 for **A**, **B** and **C**; * *p* < 0.05 for **D**). Quercetin reduced gene expression in a dose-dependent manner. Quercetin 3 µg/mL significantly decreased *il-8* (**A**, * *p* < 0.001), *cxcl-1* (**B**, * *p* < 0.005) and *il-6* (**C**, * *p* < 0.05) expression level. Quercetin 30 µg/mL was effective by reducing the expression of the four markers being tested (* *p* < 0.0001 for **A** and **B**; * *p* < 0.01 for **C** and **D**). Results are shown as the relative quantity of each gene (mean ± SEM). Statistical comparisons of the means were performed with one-way ANOVA and Sidak’s multiple comparisons test (**A**, **B**, **C** and **D**).

**Figure 4 biomolecules-09-00891-f004:**
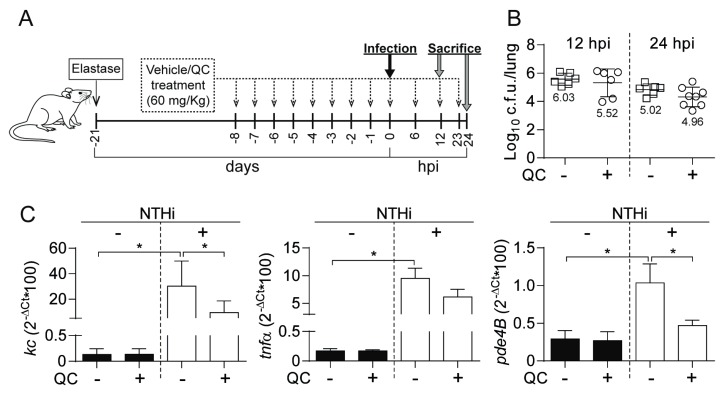
Effect of quercetin administration on bacterial loads and proinflammatory markers in emphysema mice infected by NTHi. (**A**) Experimental design: lung emphysema was induced by intratracheal instillation of pancreatic porcine elastase. Animals were infected intranasally with ~10^8^ c.f.u./mouse. Quercetin (QC, 60 mg/kg/dose) was administered orally. Controls: animals were administered vehicle solution but did not receive quercetin; (**B**) Bacterial counts were determined at 12 and 24 hpi for lung (log_10_ c.f.u./lung) samples; (**C**) Relative quantities of mouse *kc*, *tnfα* and *pde4b* mRNA were measured by RT-qPCR analysis on lung samples corresponding to non-infected untreated, non-infected quercetin treated, NTHi infected untreated, and NTHi infected quercetin treated groups. At 12 hpi, expression of the three genes was increased in infected compared to uninfected mice (* *p* < 0.005 for *kc*; * *p* < 0.0005 for *tnfα*; * *p* < 0.05 for *pde4b*), and those genes expression was lower in NTHi infected quercetin treated than in untreated mice (*kc*, * *p* < 0.05; *pde4b*, * *p* < 0.05). Results are shown as mean ± SD. Statistical comparisons of the means were performed with one-way ANOVA and Sidak’s multiple comparisons test.

**Figure 5 biomolecules-09-00891-f005:**
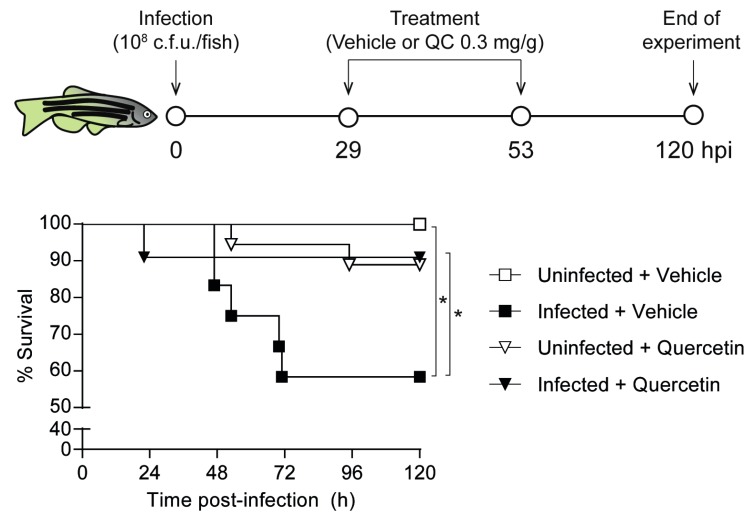
Effect of quercetin administration on zebrafish infected by NTHi. Zebrafish were infected intraperitoneally with NTHi375, ~10^8^ c.f.u./fish. When necessary, quercetin 0.3 mg/g/dose was administered intraperitoneally at 29 and 53 hpi. Non-infected groups were administered perfusion solution-DMSO (1:1) (white square) or quercetin (white triangle); infected groups were administered perfusion solution-DMSO (1:1) (black square) or quercetin (black triangle). Survival rate is reported as percentage (mean ± SD) of adult individuals survival up to 120 hpi. Survival of NTHi infected zebrafish was significantly higher in quercetin treated than in untreated animals (* *p* < 0.05). To draw and analyze the Kaplan-Meier survival curve, a Log-rank (Mantel-cox) test was used (* *p* < 0.05). Statistical comparisons between survival rates after 120 h were performed with one-way ANOVA and Sidak’s multiple comparisons test.

**Table 1 biomolecules-09-00891-t001:** The checkboard method for NTHi strain 375, when combining polyphenol:polyphenol or polyphenol:antibiotic molecules (mean ± SD).

Combination (A + B)		MIC_A_ (µg/mL)	MIC_A_ (A + B Combination)	MIC_B_ (µg/mL)	MIC_B_ (A + B Combination)	ΣFIC	Result
A	B						
Quercetin	Punicalagin	167	83.5	320	160	1	Indiferent
Quercetin	Myricetin	250 ± 118	167	800	125 ± 106	0.91 ± 0.22	Indiferent
Punicalagin	Myricetin	320	80	800	800	1.25	Indiferent
Quercetin	Azm	167	125.25 ± 59	1.5 ± 0.71	1.5 ± 0.71	2 ± 1.41	Indiferent
Myricetin	Azm	400	12.5	1.5 ± 0.71	2	1.53 ± 0.71	Indiferent
Punicalagin	Azm	320	50 ± 42.4	2	1.5 ± 0.71	0.91 ± 0.49	Indiferent
Quercetin	Amp	125 ± 59	10.45	1	1.5 ± 0.71	1.6 ± 0.75	Indiferent
Myricetin	Amp	400	800	1	2	4	Indiferent
Punicalagin	Amp	320	640	1	0.13	2.13	Indiferent

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
