# Peer review of "Preclinical Evaluation of the Antimicrobial-Immunomodulatory Dual Action of Xenohormetic Molecules against Haemophilus influenzae Respiratory Infection"

_biomolecules, 2019, doi:10.3390/biom9120891_

Round 1

Reviewer 1 Report

Dear Editor, 

I am pleased to review the assigned manuscript. Overall, a proposed review looks very good to me – Authors really did a wonderful job and presented very nice & relevant literature. I have few suggestions that can be incorporated into revised version.

Abstract looks good but just wondering if it can cover the whole theme The introduction should be better organized. Some of the sentences are not well structured, should be clarified and rewritten. It is advice to link the story in a better way in an introduction to convey a proper message to readers. Material and Methods - Well written Results and Discussion - Very clear and well organized The conclusion should be better explained and conclude your whole research.

Author Response

Dear Sir/Madam,

We appreciate very much your comments and suggestions. We have incorporated changes in the manuscript main text, that can be seen with the tracked changes tool. As requested, we have worked to improve the text in the abstract and introduction sections. We have also included a conclusion sentence at the end of the discussion section. We hope that these modifications will fulfill the requirements raised.

Reviewer 2 Report

The manuscript examines the potential of the polyphenolic molecules such as quercetin, myricetin and punicalagin against NTHi infection. The experiments are well thought of and appropriate controls are present. While this is an interesting paper and is worthy of publishing there are some minor mistakes and the manuscript needs further proofreading.

Some mistakes are pointed out as described below:

The sentence in line 65-70 is very long and difficult to understand.

Line 98-99, 355-358, 365-368 – These sentences are unclear and should be rephrased.

In line, 125-126 mention the purity of the purchased compounds.

In line 184 use 80 instead of eighty in words (be consistent).

Line 203 drug exposure was removed prior infection does not make sense and the sentence is unclear it should be rephrased.

Line 249 mention where are the experiments shown.

Give reference for the sentence, line 302.

Line 368-370 should be explained clearly. Why the cells were pretreated and how was quercetin removed?

Line 390 what is meant by intracellular bacteria?

The author uses we asked at several occasions it should be replaced by “question arises” or similar phrases similarly “In sum” should be replaced by in summary or in conclusion.

Line 512-529, this would be better placed in the introduction instead of discussion.

However, these mistakes do not detract the reader from the scientific meaning therefore; the publication of the current manuscript is recommended after correction.

Author Response

Dear Sir/Madam,

We appreciate very much your comments and suggestions. We have made changes in the manuscript main text, that can be seen with the tracked changes tool. We hope that these modifications will fulfill the requirements made.

Specific points raised:

- The sentence in line 65-70 is very long and difficult to understand. This sentence has been modified as requested.

- Line 98-99, 355-358, 365-368. These sentences are unclear and should be rephrased. These sentences have been modified as requested

- In line, 125-126 mention the purity of the purchased compounds. This information is now provided.

- In line 184 use 80 instead of eighty in words (be consistent). Changed.

- Line 203 drug exposure was removed prior infection does not make sense and the sentence is unclear it should be rephrased. This sentence has been changed.

- Line 249 mention where are the experiments shown. Information introduced.

- Give reference for the sentence, line 302. Reference introduced.

- Line 368-370 should be explained clearly. Why the cells were pretreated and how was quercetin removed? The text has been modified to make this point clearer.

- Line 390 what is meant by intracellular bacteria? H. influenzae is an intracellular facultative bacterium, able to invade human epithelial cells and locate intracellularly in subcellular compartments (doi: 10.1099/mic.0.040451-0.). In practical terms, an established manner to analyse intracellular bacteria, is the addition of gentamicin to the tissue culture medium after infection. We use a gentamicin bactericidal concentration that kills extracellular bacteria but does not permeate the host cell plasma membrane. For this reason, bacteria which have previously entered into epithelial cells, i.e. intracellular bacteria, are protected from the gentamicin killing effect, and can be analysed/quantified, etc.

- The author uses we asked at several occasions it should be replaced by “question arises” or similar phrases similarly “In sum” should be replaced by in summary or in conclusion. These modifications have been introduced.

- Line 512-529, this would be better placed in the introduction instead of discussion. Thank you for this comment. Part of this paragraph was repetitive indeed, and it is now only present in the introduction section, for clarity and to avoid repetition.

Reviewer 3 Report

Dear Authors,

The presented study describes the antibacterial (H. influenzae) and anti-inflammatory effect of  Cistus salviifolius plant extracts.

The study is well-written, well planned and the discussion follow the obtained results.

From my point of view there are no considerations concerning the quality of presented research, however there are just a few issues that should be reviewed.

It is obvious that several studies are performed on nontypable H. influenzae (NTHIi375) strain, however there is no information about the source of this particular one. How it was obtained. This information should be provided in Materials and Methods Section. I have some concerns about the assay: "NTHi serial passage in the presence of polyphenol". I believe that the aim of this assay was to find out whether the prolonged incubation lead to development of resistance. However, to do this, the MIC determination should be performed against pre-incbuated strains (at each time point), rather than OD measurement. In my opinion, these are to far-reaching conclusions. This should be discussed, and the reference to such method should be provided. The last issue, please see. Determination of plant extract and pure polyphenol antimicrobial effects. 
The proper name of the assay performed on 96-well plates is "Broth microdilution assay". Minimal Inhibitory Concentration (MIC) is a result of the procedure, not its name, so please revise. If the presented method was used (presented) in literature previously please provide specific reference.

To sum up. Generally, the highlighted issues are rather cosmetic errors. The overall quality is very good, so after their review I recommend the manuscript for publication in Biomolecules.

Author Response

Dear Sir/Madam,

We appreciate very much your comments and suggestions. We have incorporated changes in the manuscript main text, that can be seen with the tracked changes tool.

We hope that these modifications fulfill the requirements made.

Specific points:

- NTHIi375: there is no information about the source of this particular one. How it was obtained. This information should be provided in Materials and Methods Section. This information it is now provided, together with a reference.

- Assay: "NTHi serial passage in the presence of polyphenol". The aim of this assay was to find out whether the prolonged incubation lead to development of resistance. To do this, the MIC determination should be performed against pre-incubated strains (at each time point), rather than OD measurement. In my opinion, these are to far-reaching conclusions. This should be discussed, and the reference to such method should be provided.

Thank you for this comment. The text describing this assay has been modified to avoid possible overstatments and far-reaching conclusions, as indicated. This assay has been adapted from a previous study, and the reference is now included in the manuscript.

For clarity, in this experiment we incubated a defined number of bacterial CFU with a previously defined subinhibitory concentration of each polyphenol for 24 h (plate 1). The same assay was performed with a previously defined inhibitory concentration of each polyphenol. After this incubation (day 1), we measured turbidity as an indicator of bacterial growth, in turn indicative of bacteria resistance to the polyphenol. Then, we performed bacterial passages from those wells (plate 1) into new wells with the same polyphenol concentrations (plate 2) and repeated incubation and turbidity measurement. These passages were serially repeated up to day 15.

- Please see. Determination of plant extract and pure polyphenol antimicrobial effects.  The proper name of the assay performed on 96-well plates is "Broth microdilution assay". If the presented method was used in literature previously please provide specific reference.

Thank you for this comment. The text has been modified accordingly. This assay was developed for this study, by adapting a standard broth microdilution assay, and has not published elsewhere. For this reason, a specific reference cannot be provided.